# Yellow light decision based on driving style: Day or night?

**Xuan Wang[1], Yan Mao[1]\*, Jing Jing Xiong[1], Wu He[2]**

**1** School of Business, Sichuan Normal University, Chengdu, China, **2** College of Movie and Media, Sichuan Normal University, Chengdu, China

\* maoyy85@163.com

## Abstract

Drivers' driving decisions at yellow lights are an important cause of accidents at intersections. As proved by existing studies, driving style is an important basis for a driver to decide to pass a yellow light or not. This study, therefore, aims to investigate the effects of different driving styles on driving decisions at yellow lights under different lighting conditions. Specifically, 64 licensed drivers were recruited to comparative study the effects of different driving styles on the decision to pass through yellow lights under both daytime and nighttime lighting conditions using a driving simulator and a VR device. The results showed that maladjusted drivers more likely to pass the yellow light faster than adapted drivers (81.25% vs 43.75%) during both day and night. Male drivers had higher overall driving style scores than female drivers, and male drivers were faster and more likely to pass a yellow light than female drivers (56.25% vs 31.25%). This study also found that inexperienced drivers were faster and more likely to pass a yellow light than experienced drivers (50% vs 37.5%). Overall, maladjusted drivers are more likely to pass yellow lights, which can be improved and society properties by enhancing driving learning for maladjusted drivers.

**Data Availability Statement:** All relevant data are within the manuscript and its Supporting information files.

**Funding:** This work was supported by Humanities and Social Sciences projects of the Ministry of Education (18YJA760020). The authors declare that the funders had no role in study design, data

## Introduction

Nowadays, road traffic safety has become a major public concern of people safety. Because 56% of fatal road traffic accidents are usually caused by unsafe driving behaviors, which is often associated with aggressive driving style [1]. Due to the close relationship with road safety, driving style deserves deep investigation [2].

Driving style has an important influence on driver decision making. Driving style refers to a driver's habits in terms of speed selection, following distance, tendency to overtake other vehicles and violation of traffic rules [3]. It plays an important role in predicting driving behaviors and reflecting the driver's internal state [4]. Thus, it is meaningful to explore driving style. Today, there are many instruments that can be used to measure driving style, but in this study, we chose to use the Multidimensional Driving Style Inventory (MDSI). Because the MDSI has been shown to be a valid and reliable indicator for assessing driver styles [5–8]. Generally, MDSI can be divided into four driving styles through eight factors: danger, anger, anxiety, and caution [8]. In detail, dangerous driving style is the one in which the driver deliberately violates

collection and analysis, the decision to publish, or the preparation of the manuscript.

**Competing interests:** NO authors have competing interests Enter: The authors have declared that no competing interests exist.

driving regulations and is driving in pursuit of excitement, speed, or illegal overtaking; anxious driving is driving in a manner where the driver develops feelings and emotions of alertness and nervousness, accompanied by distracting behaviors; an angry driving style is the one in which the driver displays irritating, angry, and hostile attitudes and behaviors; and cautious driving style refers to safe and cautious driving behaviors [8].

The four driving styles are divided into two main categories: maladaptive driving and adaptive driving styles. Maladaptive driving styles include dangerous, anxious, and angry driving styles, while cautious driving style are adaptive driving styles [8, 9]. Drivers with different driving styles make different decisions at signal crossings. A maladjusted driver usually exhibits aggressive driving. It shows that they decided to cross the intersection and try to run the red light even though they were away from the stop line. Adaptive drivers, on the other hand, usually drive conservatively. This means that they are close to the stop line and decide to stop, even though they can safely cross the intersection [10, 11]. Although there exist many studies on driving style, there are fewer studies on drivers' driving style in different lighting conditions.

Additionally, light conditions also have an important influence on driver decisions. Firstly, drivers have different hazard perceptions under different lighting conditions. In the night scenario, the driver's hazard perception sensitivity index all but drops, and even a complete lack of awareness of the hazard occurs, which leads to an increased rate of vehicle crashes and a significant increase in the severity of injuries that do not occur in the daytime scenario [12–14]. This may be related to the psychological needs of the driver. This is because the psychological needs of drivers are higher in night scenes than in day scenes [12]. Not only that, but the psychological needs of drivers differed between driving styles as well. Therefore, it is relevant to study the influence of lighting conditions on driving style. Although there have been many studies on driving performance under different lighting conditions, such as distracted driving [15, 16], sleep driving [17, 18], driving risk [19, 20], visibility [21, 22], and visual attention [21], there is a gap in the research on the driving decisions of drivers with different driving styles under different lighting conditions.

Lastly, traffic signals at intersections have the same impact on drivers' driving decisions as stop signs do. Intersections are high-risk locations where traffic accidents occur frequently. As reported, approximately 30% of road accidents in China occur at intersections [4]. In particular, the decisions made by drivers at the start of yellow traffic lights are often critical, as inaccurate decisions win led to traffic conflicts and collisions [23]. Depending on the distance of the vehicle from the stop line, the driver is faced with two different scenarios. In the first case, the driver is in the indecision zone, while in the second case the driver is in the possible stopping zone [24]. At the onset of the yellow light signal, drivers need to decide immediately whether to stop or cross the intersection [11, 25]. However, immediate decision making by drivers is a challenge, especially in yellow light zones where drivers can neither stop safely nor cross the intersection [26]. Therefore, it is relevant to study drivers' decision making at yellow signals. Although there are many studies on signalized intersections, there is still a research gap on the decision-making behaviors of drivers with differing driving styles under different lighting conditions at yellow lights.

Early research on driver decision-making at yellow lights has focused on modelling the propensity of drivers to run yellow lights as a function of constructive driving speed, distance from the stop line, and demographic variables such as driver age and gender [11, 24, 25]. Although early studies on yellow light decision making had significant findings, there were some shortcomings that could not be addressed [10]. Today, with the advancement of technology, many scholars study driving decisions at yellow lights through improved models or functions that better reduce the likelihood of traffic accident development. For example, the

inclusion of a connected environment at signalized intersections helps drivers to make safer decisions at the onset of yellow lights [10]. In addition, distracted driving models in the form of mobile phone conversations have been constructed to investigate the impact of distracted driving on yellow light decisions [24]. While these and other related studies have confirmed the importance of drivers' yellow light decisions, it is unclear how the affective information provided by driving style affects drivers' decisions at the signal under different lighting conditions. This research gap prompted the present study.

Driving simulators plays an important role in assessing the human factors of road safety [27–29]. However, there are still limitations to the fidelity of driving simulators [21, 30]. Therefore, immersive virtual environments (IVEs) based on virtual reality (VR) technology offers an alternative approach to the study of driving behavior. Virtual reality is "a real or simulated environment in which the perceiver experiences a sense of remoteness" [31]. The behavior observed in a virtual reality environment is qualitatively comparable to that of the real world [32].

Today, IVE is an effective research tool with reasonable ecological validity to evoke realistic human driving behavior. It can control and manipulate key variables according to an experimental design and collect reliable qualitative and quantitative behavioral data [21, 33]. In this paper, we adopt a driving simulator and an IVE approach to address two questions.

In this study: Firstly, what driving decisions do drivers with different driving styles make when the traffic signal is yellow? Secondly, what driving decisions do drivers with differing driving styles make under different lighting conditions when the traffic lights are yellow?

## Materials and methods

### Participants

In this paper, 64 participants aged 19–23 years (mean age, 20.4 years (SD: 1.57 years)) were recruited for this experiment. Of these, 32 were male and 32 were female, randomly divided into four groups of 16 participants each, with equal proportions of males and females. They as have a valid driver's license with at least one year of driving experience and at least 10,000 km of driving history per year. Participants had normal or corrected to normal vision and hearing. Participants will need to sign a paper consent form before the experiment begins, complete a pre-driving questionnaire that includes questions related to demographics, driving history, and driving behavior, and perform a familiarization drive that includes interaction with traffic signal changes. Participants in the experiment were asked to obey the speed limits on the signs and drive as close to the speed limit as possible. Participants in this study were evaluated for motion sickness using an adaptation of a standard motion sickness assessment tool [34], and the NASATLX questionnaire was administered after each drive. Upon completion of the experiment, each participant was paid 50 RMB. The study was approved by the Ethics Committee of Sichuan Normal University. All experimental procedures were carried out in accordance with the Declaration of Helsinki.

Design. A 2 × 2 between-group design focused on comparing driving styles (adaptive and non-adaptive) across test variables (half of the adaptive and non-adaptive drivers were assigned to the daytime scenario and the other half to the nighttime scenario) with respect to driving decisions at yellow lights at intersections.

## Ethical approval and consent to participate

The Sichuan Normal University Committee approved the study protocol in accordance with the International Declaration of Helsinki. Oral consent was obtained from study participants and participation was entirely based on their willingness. Participants were also informed

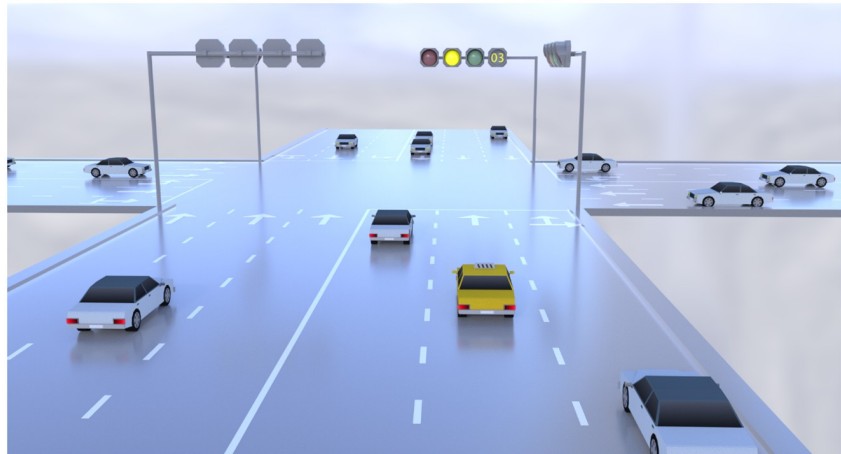

**Fig 1. Experimental scenario.**

about the purpose of the study and its procedures. Moreover, participants were informed about their right to withdraw from the study.

## Experimental equipment

### Driving simulator

The experiment was conducted using the driving simulators. Which consists of a fully functional real car and is equipped with three projectors capable of displaying 180-degree scenes. The simulator can simulate realistic driving characteristics such as acceleration, deceleration, braking, and cornering. In addition, according to Chinese road standards the driving simulator car is designed to meet the needs of the study, traffic signs and road markings. Traffic signals are distinctly placed at an intersection on an urban route. Before meeting the traffic signal, drivers drove in a simulated urban environment to familiarize themselves with the environment. When approaching an intersection with a signal, the driver must respond to the change in the traffic signal from green to yellow. The driver interacts with the traffic signal in such a way that the traffic light changes from green to yellow when the driver is 5 seconds away from the stop line [10]. Since the yellow interval between the red and green lights was set at 3 seconds (see Fig 1), it meant that participants had 2 seconds to read and interpret the information [10].

### MDSI

Based on statements of feelings, thoughts, and behaviors while driving, drivers complete a Likert scale. The scale has a total score of 6, ranging from 1 (not at all) to 6 (very much). As the Cronbach's alpha was reasonable for the four dimensions (0.82 for dangerous driving, 0.82 for anxious driving, 0.77 for angry driving, and 0.70 for cautious driving), each driver's responses to the relevant scales were averaged to produce scores of each of the four driving styles, with higher scores indicating higher levels of that style.

### Driver demographics

Driving gender and driving experience have a significant impact on driving behaviors. Drivers of different genders exhibit different driving behaviors. Male drivers tend to drive faster and are more prone to reckless driving, while female drivers tend to choose to drive more

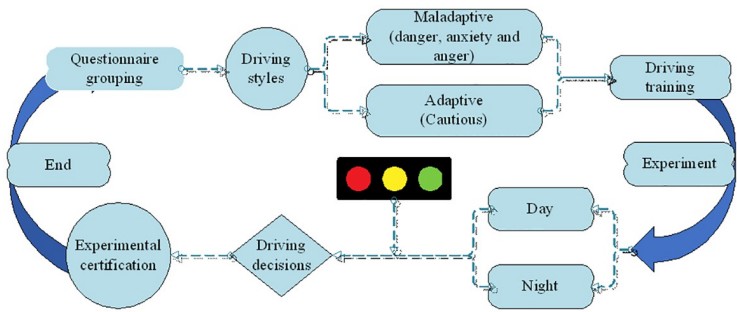

**Fig 2. Experimental procedures.**

cautiously. Driving experience also has a varying degree of influence on driving behaviors. Experienced drivers tend to choose more dangerous driving styles, while inexperienced drivers tend to be overly cautious and cause anxiety. This paper therefore investigates the differences between the driving styles of drivers in different light conditions by selecting driving gender and driving experience as demographic variables to fill in the gaps in the research. To prevent confounding of demographic variables by driving style and light conditions, drivers of the same driving gender and number of driving experiences were selected for the study.

## Experimental procedure

The experiment recruited 64 subjects and each subject was given two opportunities to experiment. The flow of the experiment was given as follows (as shown in Fig 2).

1. Participants arrived at the laboratory to sign an informed consent form for the experiment and fill out relevant personal information (age, gender, driving experience, driving history, and questions related to driving behavior) and perform a familiarization drive that included interaction with traffic signal changes.

2. The drivers were differentiated by using the MDSI questionnaire, and subjects were divided into four groups: daytime non-adapted, daytime adapted, nighttime adapted and nighttime non-adapted. And subjects were assigned with individual tasks. Firstly, training in the operation of the driving simulator was carried out; secondly, after the realism of the driving experience in the simulator [35], the road environment was simulated as naturally and as detailed as possible. The scenario was consistent between the driving simulator and the VR system. Which training was completed a formal driving simulation was carried out by using the driving simulator.

3. After informing, subjects were placed on the driving simulator for training and performed a familiarization drive, including interactions with traffic signal changes, and 15 minutes later, a formal driving simulation.

4. Before starting the experimental driving, participants received the following instructions: "Drive as usual until you pass the intersection. You have the option to withdraw from the experiment at any time for any reason." For the formal driving, the average participant was required to react to the traffic light ahead changing from green to yellow in a daytime scenario. The other half of the participants performed the same experiment in a nighttime scenario.

5. After the simulated driving is completed, an equal number of participants will be randomly selected from each of the 4 groups to be certified for the VR experiment.

6. After completing the test, each participant was asked to answer a post-test questionnaire. The questionnaire included feedback and thoughts on the driving experience and the driving simulator itself, as well as an assessment of the effect of the tool on the participant's motion sickness using the NASATLX questionnaire.

## Data processing

Driver decisions were derived from driving simulator data. The explanatory variables are divided into traffic operation variables, driver demographics, and driving conditions.

Traffic operation variables include driving speed, distance to the stop line at the start of the yellow light, acceleration noise (or change) before the start of the yellow light. Driver demographics include age, gender, driving experience, license type, and education. Driving condition variables had two categories: daytime and nighttime. The study used analysis of variance to test the experimental hypothesis. Namely, that driving style is significantly associated with yellow light driving decisions. Illumination conditions influence driving decisions on driving styles.

Sixty-four participants drove twice, resulting in 128 decisions at the start of the yellow interval at signalized intersections. Each driver interacted with the traffic signal twice in repeated driving, resulting in a data set. Summary statistics for the explanatory and response variables (Table 1). Of the 128 encounters with the yellow light, 18 drivers decided to cross the intersection in the daytime scenario, and 38 drivers crossed the intersection in the nighttime scenario. The average driving speeds for the daytime and nighttime light conditions were 11.97m/s (SD:2.26m/s) and 10.69m/s (SD:2.16m/s), respectively. Acceleration noise, as the standard deviation of roadway acceleration before the onset of yellow light, is considered an indicator of reckless driving [36], with values of 0.38 m/s2(SD (0.04) m/s2) and 0.37 m/s2(SD (0.04) m/s2) for daytime and nighttime ambient driving conditions, respectively.

## Results

### Decision trees

A decision tree is constructed from the available data to classify the discrete outcomes of drivers' stopping or passing at the start of a yellow indication at a signalized intersection. The dependent variable is the binary outcome in the decision tree (i.e., the driver's decision to stop at a traffic signal intersection), and the input variables are driving conditions, driver's gender,

**Table 1. Summary statistics of operation and response data for each driving scenario.**

| variable | Variable Description | Average(SD) | | Count (percentage) | |
|---|---|---|---|---|---|
| | | **Day** | **Night** | **Day** | **Night** |
| Driving conditions | | | | | |
| Day | Driving in daytime scenarios | - | - | 32 | 100 |
| Night | Driving at night scenario | - | - | 32 | 100 |
| Traffic operation variables speed | Speed of driver at start of yellow light(m/s) | 11.97(2.26) | 10.69 (2.16) | - | - |
| Acceleration noise (or variation) | Standard deviation of driver's acceleration and deceleration before the start of the red light(m/s^2) | 0.38(0.04) | 0.37(0.04) | - | - |
| length | Distance from the stop line at the start of the yellow light | 37.66(4.29) | 36.75 (4.20) | - | - |
| Response Variables decision-making | The driver decided to proceed through a yellow light | | | 10(47.75) | 18 |

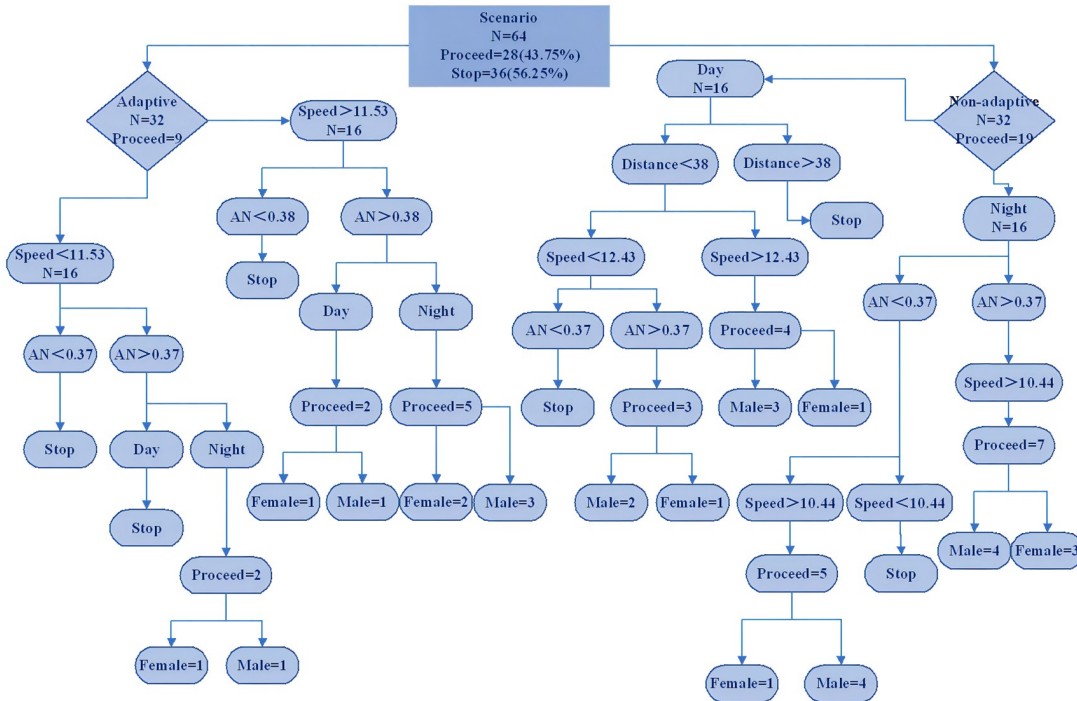

**Fig 3. Decision trees.** Schematic diagram of the decision tree for the stop/pass decision model. Note that the numbers in the circles denote interaction terms; distance, AN, and speed denote the distance to the stop line at the start of the yellow light, acceleration noise (m/s2), respectively.

traffic operation variables, and driver demographics provided as possible explanatory variables in the decision tree. Fig 3 shows the decision tree diagram for the stop/pass decision model. Nodes are identified by selecting the option that provides the highest information gain. The decision tree classifies driver decisions by dividing the data into 52 smaller homogeneous groups, and their corresponding statistics are given within each node. The total number of nodes reached (N) and the number of stops or passes at the intersection for that node are presented in the Fig 3. For example, terminal node 3 indicates that about 12.5% of drivers choose to pass the yellow light at a speed of 11.53 m/s and an acceleration of 0.37 m/s for 2 the adapted drivers. Similarly, terminal node 15 implies that drivers with a distance greater than 38m are likely to stop in the daytime scenario. In terminal node 19, 57.14% of male drivers with an acceleration noise of 0.37 m/s2 and a speed greater than 10.44 m/s2 choose to pass the yellow light in the night scenario.

## Driving style

In this paper, driving style is scored by the MDSI driving questionnaire, and the results of the MDSI questionnaire are divided according to gender. This paper examines the differences between driving style and gender by means of t-tests. The mean and SD scores are shown in Table 2. The results show that there is a significant difference between driving styles and gender. This result shows that adaptive driving style scores are overall higher than non-adaptive driving scores, and males score higher than females. Also, the results showed that males scored higher than females on anxious and cautious driving styles. Females scored higher than males on dangerous and angry driving styles.

**Table 2. Driving style score according to gender.**

| Driving style | MDSI factors | Men | Women |
|---|---|---|---|
| Non-adaptive | Dissociative and Anxious | | |
| | Mean | 2.31 | 2.23 |
| | S.D. | 0.77 | 0.25 |
| | Risky and Angry | | |
| | Mean | 2.04 | 2.05 |
| | S.D. | 0.77 | 0.74 |
| | High-velocity and Distress reduction | | |
| | Mean | 2.67 | 2.43 |
| | S.D. | 1.01 | 0.89 |
| Adaptive | Patient and Careful | | |
| | Mean | 4.65 | 4.57 |
| | S.D. | 0.78 | 0.69 |

## Different light conditions

This paper conducts a 2 (driving styles: maladaptive, adaptive) X 2 (lighting conditions: day, night) simulation experiment on drivers' driving decisions. An analysis of variance (ANOVA) was conducted on the results collected from the experiments. The results showed that light conditions had a significant effect on driving speed ($F_{(1,62)} = 10.33$, $p < 0.05$). Drivers in the daytime scenario were slower than those in the nighttime scenario (11.97m/s vs 10.69m/s). Light conditions also had a significant effect on acceleration noise ($F_{(1,62)} = 0.18$, $p < 0.05$) and distance ($F_{(1,62)} = 0.67$, $p < 0.05$). Driver acceleration noise was lower in the daytime scenario than in the nighttime scenario (0.374 vs 0.375) and driver distance from the yellow light was greater in the daytime scenario than in the nighttime scenario (37.66 vs 36.75). The results also revealed that the drivers in the night scenario had a higher probability of proceeding through a yellow light than the drivers in the day scenario (59.375% vs 28.125%).

## Discussion

### Driving decisions under different light conditions

Driver behavior at signalized intersections is considered critical because of its direct impact on traffic safety [37]. And different light conditions may lead to deviations in driving behavior. Such deviations can greatly increase the incidence of traffic accidents. For this reason, VR simulations of lighting scenarios provide realistic and important information for research and are expected to reduce the incidence of deviations.

Fig 4 shows that the probability of a driver choosing to pass a yellow light is lower in the daytime situation than in the nighttime situation (31.25% vs 56.25%). This increased probability can be explained by the fact that the driving behavior of the sampled drivers in the nighttime scenario may lead to an increased risk. This result can be attributed to the fact that drivers have different visibility of the road under different lighting conditions [38]. In low visibility conditions, accidents are more likely to occur due to difficulties in recognizing conditions on the road, especially in encounters with pedestrians [39, 40]. Conversely, in high visibility conditions, drivers can make the right decision in time to avoid an accident when encountering road conditions.

Fig 5 shows that different lighting conditions have a significant effect on driving speed ($F_{(1,62)} = 10.33$, $p < 0.05$). The driver's speed in the daytime scenario was lower than the driver's

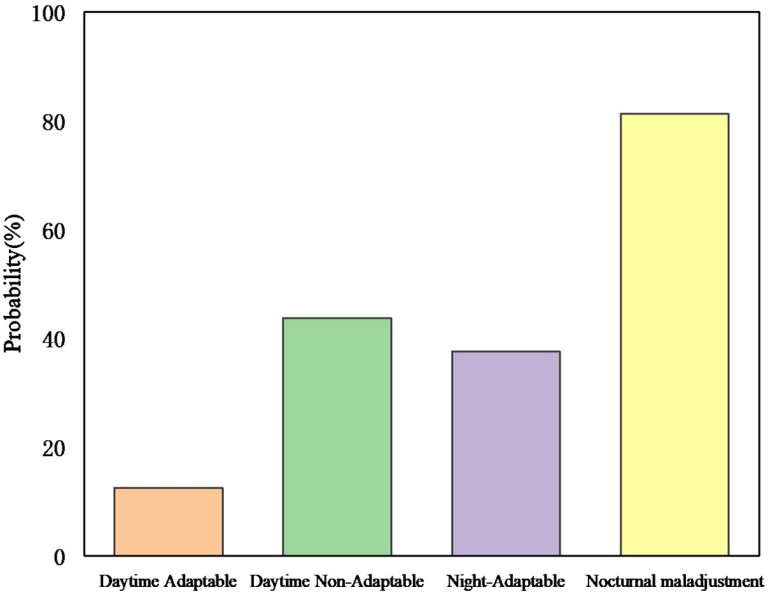

**Fig 4. Probability of driver passing yellow light under interaction.**

speed in the nighttime scenario. This result can be explained by the fact that drivers are more likely to control their emotions in the daytime scenario and tend to choose to follow driving safety and traffic regulations, so they choose to reduce their speed when encountering yellow lights. On contrasted the night scenario, drivers tend to disregard driving safety norms and perceive high speeds as safer due to changes in the environment and visibility, so they choose to overtake at high speeds in the yellow light scenario. Fig 5 also shows that different lighting conditions have a significant effect on driver driving acceleration ($F_{(1,62)} = 0.18$, $p < 0.05$).

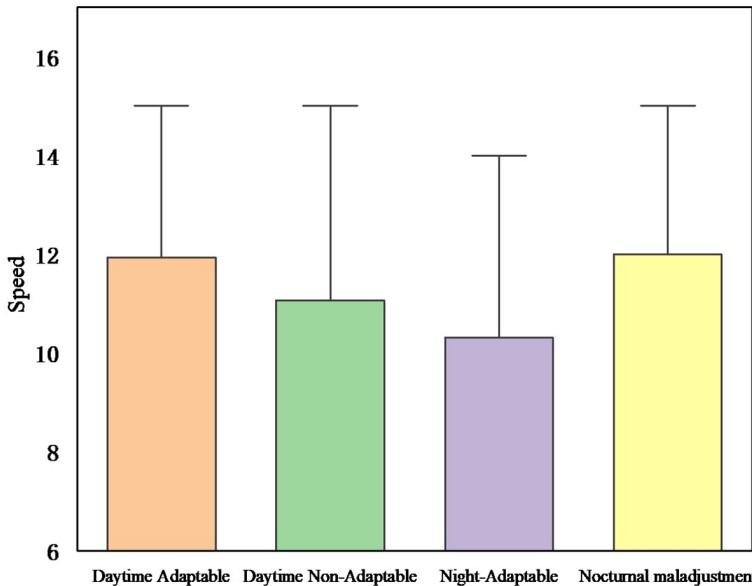

**Fig 5. Speed and acceleration of driver through yellow light under interaction.**

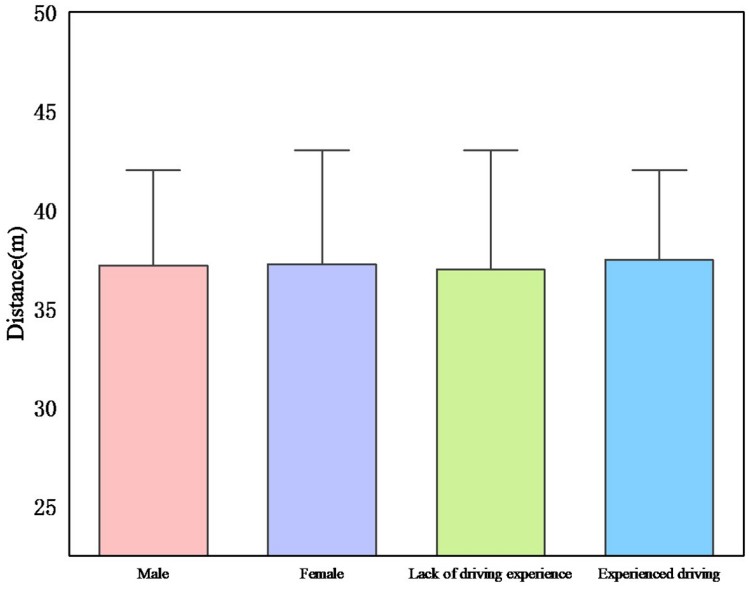

**Fig 6. Distance(a).**

Driver driving acceleration at night was greater than the driver acceleration during the day, as different light conditions determined driver speed selection. In particular, the difference between the acceleration chosen by maladjusted drivers at night and during the day is greater, which may be related to changes in driver mood.

As can be seen in Fig 6, lighting conditions had a significant effect on driving distance (F $(1,62) = 0.67$, $p<0.05$). The distance between the driver and the stop line in the night scene is smaller than that in the daytime scene. This result can be explained by the fact that the driver's distance from the stop line was negatively related to the driver's driving decision to pass the yellow light. The shorter the distance between the driver and the stop line, the more likely the driver is to make the decision to pass the yellow light. In the nighttime environment, drivers tend to increase their speed, so the shorter the distance to the stop line, the higher the probability of passing the yellow light.

Overall, the findings suggest that drivers tend to choose higher speeds to pass the yellow light in the night scenario than in the day scenario, while the distance to the stop line is shorter and acceleration is higher.

## Driving decisions under different driving styles

The results of the study show that driving style has a significant influence on the driver's driving decision to choose to pass a yellow light. This result suggests that driving style has an important basis for drivers to make driving decisions. Driving style predicts driver behavior on the road, which in turn reduces the incidence of traffic accidents.

From Fig 4, it can be concluded that non-adaptive drivers are more likely to choose to pass a yellow light than adaptive drivers (59.375% vs 28.125%). This can be attributed to the fact that non-adaptive drivers are more likely to make dangerous decisions and tend to violate safe driving norms by passing yellow lights quickly. Adaptive drivers are more likely to make rational decisions due to their emotional stability and tend to follow safe driving guidelines by stopping and waiting at the stop line. This result further emphasizes the importance of driving style in predicting driving behavior.

In the literature, drivers' speed is repeatedly cited as a contributing factor in their decision to start at a yellow light. In general, drivers usually choose a faster speed to run a yellow light. To test whether our data would yield factually correct results, an ANOVA was chosen for testing. The results showed that driving style had a significant effect on drivers' driving speeds ($F_{(1,62)} = 6.83$, $p<0.05$). Drivers with different driving styles chose different driving speeds. The results also suggest that when non-adaptive drivers drive at higher speeds, they tend to choose a more dangerous route (frequent lane changes) to get through yellow lights. This result can be explained by the fact that non-adaptive drivers tend to perceive themselves as skilled drivers and that higher speeds produce a sense of excitement, which makes non-adaptive drivers feel happy emotions. In contrast, adaptive drivers are ruled by caution and prudence when driving and tend to drive within the prescribed speed limit. As a result, non-adaptive drivers had significantly higher speeds than adaptive drivers (11.53 m/s vs 10.13 m/s). Similarly, as acceleration noise increased ($F = 0.27$, $p<0.05$) and distance decreased ($F = 0.25$, $p<0.05$), the probability of passing a yellow light was significantly higher for non-adaptive drivers compared to adaptive drivers. A similar explanation can be made for the relationship with the probability of passing a yellow light, i.e., the driver's driving style had a significant effect on the change in acceleration, with a positive correlation ($F = 0.27$, $p<0.05$). Therefore, to reduce the probability of yellow light jumping by maladjusted drivers, relevant driving regulations should be developed to regulate the behavior of this group of drivers and reduce the probability of traffic accidents.

## The impact of interactions on driving decisions

The presence of interaction effects suggests a degree of complexity between driver decision-making at yellow lights and data on speed, distance to the stop line, acceleration, lighting conditions, and driving style. These relationships are further complicated by the effect of driving style on drivers' speed choices at signalized intersections. To test for differential risk, the probability of a driver passing a yellow light was plotted as shown in Fig 4. The results show that the interaction between driving style and lighting conditions had a significant effect ($p<0.05$) on the driver's driving decision to choose to pass the yellow light. In daytime conditions, maladjusted drivers were more likely to choose to pass the yellow light than adapted drivers (43.75% vs 12.5%) and in nighttime conditions (81.25% vs 43.75%).

The results also suggest that the interaction between lighting conditions and driving style has a negative impact on driving decisions. The effects of different lighting conditions were different for drivers with different driving styles. Firstly, for non-adaptive drivers, adequate lighting mitigated the behavior of passing yellow lights and reduced the likelihood of injury or death (43.75% vs 81.25%). Secondly, for adaptive drivers, the effect of light conditions on passing yellow lights was significant ($F_{(1, 31)} = 0.04$, $p<0.05$), indicating that drivers were more likely to choose to pass yellow lights in nighttime conditions (12.5% vs 32.5%).

## The impact of driver demographics on the probability of running yellow lights (Table 3)

**Driver's gender.** Fig 7 shows the probability of passing the yellow light among drivers. The probability of passing a yellow light increases with speed for all drivers in both conditions, and males are shorter from the stop line than females (Fig 8). The results demonstrate that the probability of passing a yellow light is higher in night driving conditions and that males have a better chance of passing a yellow light than females. Because the distance is shorter, and the probability of passing is greater. For example, the probability of running a yellow light for male drivers in daytime driving conditions is 37.5%, while the probability of passing a yellow

**Table 3. Driving style score according to gender.**

| Driving demographics | Variable Description | Driving Speed(m/s) | Count(percentage) Day Night |
|---|---|---|---|
| Driving gender | Male | 11.50 | 6(56.25)12 |
| | Female | 11.16 | 3(31.25)7 |
| Driving experience | lack | 11.68 | 5(50.00)11 |
| | plentiful | 11.00 | 4(37.50)8 |

light for male drivers in nighttime conditions is 75%, which indicates that the probability of passing a yellow light is reduced by half. This reduction in probability can be attributed to the higher driver visibility in daytime scenarios [38], which is in line with studies that found that women are more hesitant than men to run red lights when drivers decide to do so [41].

The study found that the night environment provides better environmental conditions for male drivers to pass compared to female drivers. It is because males have more risk-taking behaviors than females and are more at risk of car accidents and are more likely to violate traffic rules [42]. For example, compared to the daytime scenario, the probability of passing for female drivers is about 18.75%, while the probability of passing for male drivers is about 37.5%, which means that male drivers seem to prefer to pass the yellow light, while female drivers prefer to wait at the stop line at night [37]. This is because males are more likely to make dangerous decisions than females and believe that they are less likely to be in a car accident. Females, on the other hand, are conservative and adhere to safe driving practices. And Fig 9 indicates that there is a gender difference in driving speed, with male drivers driving at a higher speed than female drivers. This result also confirms the influence of light conditions and driving style on driving decisions. That is, daytime adaptive drivers tend to drive at lower speeds and avoid running yellow lights.

**Driver experience.** Figs 7–9 show the probability of passing a yellow light for all driving experience groups. It can be observed that in both cases, the probability of passing the yellow

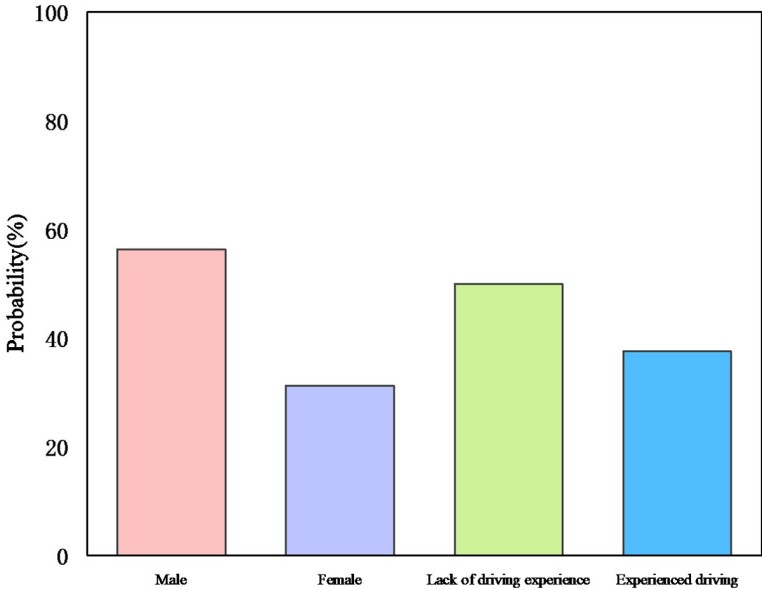

**Fig 7. Effect of driving demographics on the probability of passing a yellow light.**

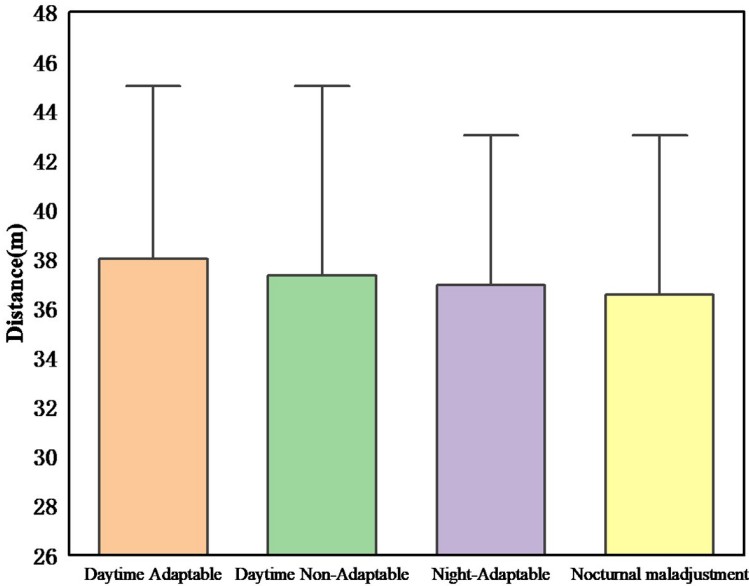

**Fig 8. Distance(b).**

light increases with increasing speed for all drivers, while in the night scenario driving conditions, the probability of passing the yellow light is higher for experienced drivers and the driving distance from the stop line is also greater than for inexperienced drivers. This is because they tend to cross the intersection by increasing their speed at the start of the yellow light. However, as found in this study, the occurrence of this risky behavior can be reduced by predictive information on driving style.

Inexperienced drivers appeared to have a lower propensity to pass through yellow lights. For example, the probability of passing the yellow light for inexperienced drivers in the

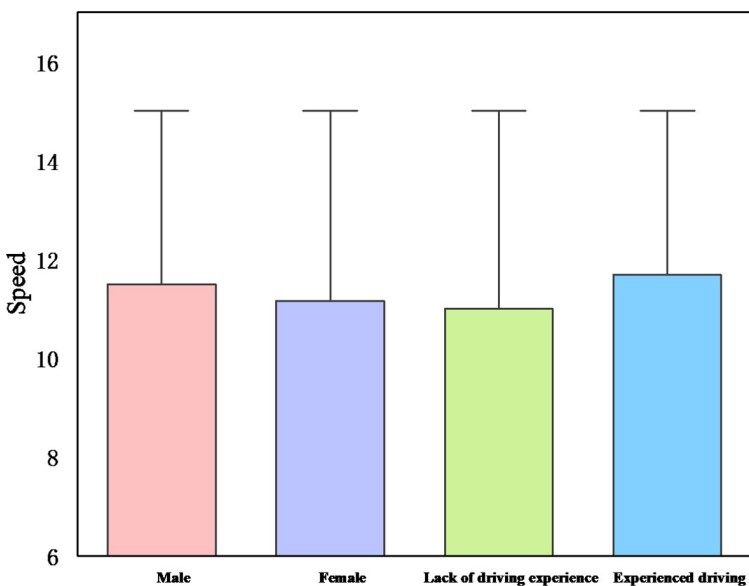

**Fig 9. Effect of demographics on speed.**

daytime scenario condition is 31.25%, while that for experienced drivers in the same speed condition is 25%, which indicates a decrease in the probability of passing the yellow light. This decrease in probability can be attributed to the slower driving speed of inexperienced drivers in the daytime scenario. Inexperienced drivers are normally considered dangerous because they tend to cross the intersection at the beginning of the yellow light by increasing their speed or running the red-light violation, while drivers with more driving experience have more experience in processing information, deciding, and taking safe actions to avoid potentially dangerous events. Whereas, in the night scenario, the probability of passing the yellow light for inexperienced drivers is 50%, while that for experienced drivers under the same speed condition is 68.75%, which indicates an increase in the probability of passing the yellow light. This increase in probability can be attributed to the fact that experienced drivers choose to drive faster in the nighttime environment. More specifically, experienced drivers are more likely to choose to drive at high speeds compared to the speed of the daytime scenario. Because experienced drivers believe that they are proficient in driving operations, have good road perception, can make correct decisions, and trust their skills, no dangerous accidents will occur.

## Conclusions and limitations

In this paper, we investigate the influence of a driver's driving style on driving decisions at intersections with yellow lights by comparing different lighting conditions. Based on the results obtained, some remarks can be listed as follows:

1. The probability of a driver deciding to wait at the stop line in the daytime scenario is greater than the probability of waiting in the nighttime scenario. (71.875% greater than 40.625%)

2. Non-adaptive drivers are more likely to choose to pass a yellow light rather than to wait at the stop line. (62.5%)

3. During the day, adaptive drivers tended to wait at the stop line, while non-adaptive drivers tended to pass the yellow light; during the night, non-adaptive drivers had a much higher probability of passing the yellow light than during the day, and adaptive drivers had a much higher probability of passing the yellow light.

4. Male drivers had higher overall driving style scores than female drivers, and male drivers were more likely to pass a yellow light than female drivers with higher driving speeds, higher acceleration, and smaller distances (56.25% vs 31.25%).

5. Drivers with a lack of driving experience were more likely to pass a yellow light than experienced drivers with higher driving speeds, higher acceleration, and smaller distances (50% vs 37.5%).

The purpose of this study is to make recommendations by comparing daytime and nighttime drivers' decisions to pass the yellow light. Since the probability of passing a yellow light at night is significantly higher than that during the day, it is recommended that:

1. Add education on traffic lights to the driving test, especially the 3-second yellow light, so that the driver do not jump the yellow light.

2. It is possible to rationalize the length of traffic signals during the day and at night, as well as to increase the duration of traffic signals at night as needed.

3. For maladjusted drivers, they can learn about traffic signals and watch videos of accidents related to yellow light jumping online, then they can realize the seriousness of the incident

and achieve a reduction in yellow light jumping; for adapted drivers, they can learn to prevent yellow light jumping through videos.

Although this paper investigates the driving decisions of drivers with different driving styles under yellow lights based on different lighting conditions, there are same limitations:

1. This article is more general in its classification of driving styles. Under maladaptive driving styles, there are three different driving styles: dangerous, anxious, and anger driving styles. Whether these three driving styles are still used with the conclusions of this paper is worthy future research.

2. This paper only considers the yellow light decision in daytime and nighttime scenarios, and it is unclear whether it is used in other scenarios, especially in bad weather.

3. The yellow interval studied in this paper is 3 seconds, but a 3-second yellow interval is not sufficient [43]. Future studies can be conducted on different yellow light intervals and different intersection types.

## Supporting information

**S1 Data.**
(XLSX)

## Author Contributions

**Conceptualization:** Xuan Wang, Yan Mao, Jing Jing Xiong, Wu He.

**Data curation:** Xuan Wang, Wu He.

**Formal analysis:** Xuan Wang.

**Funding acquisition:** Yan Mao, Wu He.

**Project administration:** Yan Mao.

**Supervision:** Yan Mao.

**Writing – original draft:** Xuan Wang.

**Writing – review & editing:** Xuan Wang, Yan Mao.

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
