## [Decision Letter · Decision Letter 0]

11 Jan 2022

PONE-D-21-39684Yellow light decision based on driving style: day or night?PLOS ONE

Dear Dr. Mao,

Thank you for submitting your manuscript to PLOS ONE. After careful consideration, we feel that it has merit but does not fully meet PLOS ONE’s publication criteria as it currently stands. Therefore, we invite you to submit a revised version of the manuscript that addresses the points raised during the review process.

We look forward to receiving your revised manuscript.

Kind regards,

Feng Chen

Academic Editor

PLOS ONE

Journal Requirements:

3. PLOS ONE does not copy edit accepted manuscripts (https://journals.plos.org/plosone/s/criteria-for-publication#loc-5). To that effect, please ensure that your submission is free of typos and grammatical errors. 

"NO - Include this sentence at the end of your statement: The funders had no role in study design, data collection and analysis, decision to publish, or preparation of the manuscript."

7.Please review your reference list to ensure that it is complete and correct. If you have cited papers that have been retracted, please include the rationale for doing so in the manuscript text, or remove these references and replace them with relevant current references. Any changes to the reference list should be mentioned in the rebuttal letter that accompanies your revised manuscript. If you need to cite a retracted article, indicate the article’s retracted status in the References list and also include a citation and full reference for the retraction notice.

Reviewers' comments:

Reviewer's Responses to Questions

**Comments to the Author**

1. Is the manuscript technically sound, and do the data support the conclusions?

Reviewer #1: Yes

Reviewer #2: Yes

2. Has the statistical analysis been performed appropriately and rigorously? 

Reviewer #1: Yes

Reviewer #2: Yes

3. Have the authors made all data underlying the findings in their manuscript fully available?

Reviewer #1: Yes

Reviewer #2: Yes

4. Is the manuscript presented in an intelligible fashion and written in standard English?

Reviewer #1: Yes

Reviewer #2: Yes

5. Review Comments to the Author

Reviewer #1: The authors propose that driving style and lighting conditions have an impact on the driver's decision to drive at yellow lights. The study was conducted using a driving simulator and a VR device, which has led to new technological developments to address the driving aspect of the study. The study shows that maladjusted drivers are more likely to run yellow lights and point to relevant solutions that are relevant to solving the problem of yellow light running.

The paper is interesting and should be useful to some practitioners and researchers in the field. However, the main drawback is that the contribution of this work is not specific enough, examining only the two main categories of driving styles without studying them separately in detail.

I have the following suggestions for how the authors could improve the submission:

1）Although the text is generally well written, there are errors in grammar and misspellings that make comprehension difficult. For example, on page 3 in the paragraph: “modelling- modeling”. For example, in this paper, driving styles or driving style? I strongly recommend that the author revise the text to address these errors prior to publication.

2）The traffic running variables in the article include driving speed, distance to the stop line at the start of the yellow light, acceleration noise (or change) before the start of the yellow light, time, and minimum speed. However, the two variables, time, and minimum speed are not shown in the table and decision tree. Are these two variables necessary or unnecessary? If they are not necessary, please fix them, and if they are necessary, please add the relevant data information.

3）The ideas presented are already known, so probably you need to improve the discussion so you can add some novelty to this paper.

4）The experimental design was demonstrated clearly in the paper. The authors tried to analyze the effects of different demographic information but is it confounded with other variables during group assignment?

5）The authors adopted ANOVA for statistical analysis but lack interpretation. Please add the relevant information in Data Processing.

6）How were the driving styles manipulated? Please add relevant information in the Experiments section.

Reviewer #2: The authors use a driving simulator and a VR device to study the influence of driving style and lighting conditions on drivers' yellow light decisions. The results show that maladjusted drivers are more likely to run yellow lights. The results of this study are relevant to solving practical problems. The manuscript is well structured and readable. Below I point out some issues I consider the authors need to address for the manuscript to be accepted.

1. There are grammatical errors. Please have a native speaker proofread the manuscript and correct the English grammar and exposition accordingly.

2. The bar graphs in the article are not sufficiently aesthetically pleasing. The visibility and accuracy of the bar graphs are low, so please use professional software to create them.

3. An explanation of the demographic variables and why driving experience and driving gender were chosen as demographic variables should be added. Without the explanation of the demographic variables, the significance of the variables is not clear. You can provide details in the Experiments or Results section.

4. The introduction section is detailed, but the logic between sentences needs strengthening. A systematic and complete revision of the relationships between statements is recommended.

5. There are problems with the format of the references. For example, “Lee, & John. Handbook of driving simulation for engineering, medicine, and psychology. Crc Press. 2011.”

6. PLOS authors have the option to publish the peer review history of their article (what does this mean?). If published, this will include your full peer review and any attached files.

Reviewer #1: No

Reviewer #2: No

---

## [Author Response · Author response to Decision Letter 0]

9 Feb 2022

Editor: 

1. Please ensure that your manuscript meets PLOS ONE's style requirements, including those for file naming. The PLOS ONE style templates can be found at https://journals.plos.org/plosone/s/file?id=wjVg/PLOSOne_formatting_sample_main_body.pdf and https://journals.plos.org/plosone/s/fileid=ba62/PLOSOne_formatting_sample_title_authors_affilia tions.pdf 

Thanks to the editor's valuable comments, we have made corrections to the layout of the manuscript and the naming of the files. 

The original manuscript sentence was “Participants were required to sign an informed consent form prior to the start of the experiment”, Following the editor’s suggestion, we updated the sentence as “Participants will need to sign a paper consent form before the experiment begins”. We have also included a sub-section on “Ethical approval and consent to participate” in the methods section, which provides detailed information on the ethical statement for the study.

“Ethical approval and consent to participate The Sichuan Normal University Committee approved the study protocol in accordance with the International Declaration of Helsinki. Oral consent was obtained from study participants and participation was entirely based on their willingness. Participants were also informed about the purpose of the study and its procedures. Moreover, participants were informed about their right to withdraw from the study。” 

3. PLOS ONE does not copy edit accepted manuscripts (https://journals.plos.org/plosone/s/criteria- for-publication#loc-5). To that effect, please ensure that your submission is free of typos and grammatical errors. 

We sincerely thank the editor for raising grammatical issues. We have thoroughly proofread the entire manuscript, as detailed below:

Page 1, Line 6, 8, 12 :driving styles----driving style 

Page 1, Line 7 : driving styles deserve--driving style deserves 

Page 1, Line 15 :divide--divided 

Page 2, Line 31, 32 : driving styles----driving style 

Page 3, Line 77 : play---- plays 

Page 3, Line 101, 103 : included---- includes 

Page 3, Line 111 : design---- design 

Page 6, Line 186-187 : variables divided ---- variables are divided 

Page 7, Line 225, 226, 228 :driving styles----driving style 

Page 7, Line 227: to the gender ---- to gender 

Page 8, Line 239 : the drivers----those 

Page 8, Line 267 : therefore----so they 

Page 11, Line 384: as dangerous drivers ---- dangerous 

Page 11, Line 395-396: and they trust their skills and believe that no dangerous accidents will occur.---- and trust their skills, no dangerous accidents will occur. 

Page 11, Line 398, 410 :driving styles----driving style 

Page 12, Line 430 : the maladaptive driving styles ---- maladaptive driving styles 

4. Thank you for stating the following financial disclosure: "NO - Include this sentence at the end of your statement: The funders had no role in study design, data collection and analysis, decision to publish, or preparation of the manuscript." At this time, please address the following queries: a) Please clarify the sources of funding (financial or material support) for your study. List the grants or organizations that supported your study, including funding received from your institution. b) State what role the funders took in the study. If the funders had no role in your study, please state: “The funders had no role in study design, data collection and analysis, decision to publish, or preparation of the manuscript.” c) If any authors received a salary from any of your funders, please state which authors and which funders. d) If you did not receive any funding for this study, please state: “The authors received no specific funding for this work.” Please include your amended statements within your cover letter; we will change the online submission form on your behalf. 

Following the editor’s suggestion, we have included detailed information in the cover letter.

“This work was supported by Humanities and Social Sciences projects of the Ministry of Education(18YJA760020).The authors declare that the funders had no role in study design, data collection and analysis, decision to publish, or preparation of the manuscript.” 

"Upon re-submitting your revised manuscript, please upload your study’s minimal underlying data set as either Supporting Information files or to a stable, public repository and include the relevant URLs, DOIs, or accession numbers within your revised cover letter. For a list of acceptable repositories, please see http://journals.plos.org/plosone/s/data-availability#loc-recommended- repositories. Any potentially identifying patient information must be fully anonymized. 

Important: If there are ethical or legal restrictions to sharing your data publicly, please explain these restrictions in detail. Please see our guidelines for more information on what we consider unacceptable restrictions to publicly sharing data: http://journals.plos.org/plosone/s/data- availability#loc-unacceptable-data-access-restrictions. Note that it is not acceptable for the authors to be the sole named individuals responsible for ensuring data access. 

As suggested by the editor, we upload the relevant information about the data in the support information under the name "Data". 

6. PLOS requires an ORCID iD for the corresponding author in Editorial Manager on papers submitted after December 6th, 2016. Please ensure that you have an ORCID iD and that it is validated in Editorial Manager. To do this, go to ‘Update my Information’ (in the upper left-hand corner of the main menu), and click on the Fetch/Validate link next to the ORCID field. This will take you to the ORCID site and allow you to create a new iD or authenticate a pre-existing iD in Editorial Manager. Please see the following video for instructions on linking an ORCID iD to your Editorial Manager account: https://www.youtube.com/watch?v=_xcclfuvtxQ.

Following the editor's advice, we registered an ORID account under the following name: maoyy85@163.com. 

7.Please review your reference list to ensure that it is complete and correct. If you have cited papers that have been retracted, please include the rationale for doing so in the manuscript text, or remove these references and replace them with relevant current references. Any changes to the reference list should be mentioned in the rebuttal letter that accompanies your revised manuscript. If you need to cite a retracted article, indicate the article’s retracted status in the References list and also include a citation and full reference for the retraction notice. 

We sincerely thank the editors for raising the issue of references. We have thoroughly proofread the entire manuscript, as detailed below: 

“1. Ali, Y. and Haque, M. M. and Zheng, Z. and Bliemer, Mcj. Stop or go decisions at the onset of yellow light in a connected environment: A hybrid approach of decision tree and panel mixed logit model. Analytic Methods in Accident Research. 2021;31(7–8):100165. 

2. Yasir Ali and Zuduo Zheng and Md. Mazharul Haque and Mehmet Yildirimoglu and Simon Washington Understanding the discretionary lane-changing behaviour in the connected environment. Accident Analysis & Prevention. 2020;137:105463. 

3. Almutairi, O. and Wei, H. Effect of speed/red-light cameras and traffic signal countdown timers on dilemma zone determination at pre-timed signalized intersections. Accident Analysis & Prevention. 2021;154:106076. 

4. Asadamraji, Morteza and Saffarzadeh, Mahmood and Ross, Veerle and Borujerdian, Aminmirza and Sheikholeslami, Sina. A novel driver hazard perception sensitivity model based on drivers’ characteristics: A simulator study. Traffic Injury Prevention. 2019;20(5):492-497. 

5. Francesco Bella. Driving simulator for speed research on two-lane rural roads. Accident Analysis & Prevention. 2008;40(3):1078-1087. 

6. Nipjyoti Bharadwaj and Praveen Edara and Carlos Sun. Sleep disorders and risk of traffic crashes: A naturalistic driving study analysis. Safety Science. 2021;140:105295. 

7. Alex A. Black and Rebecca Duff and Madeline Hutchinson and Ingrid Ng and Kirby Phillips and Katelyn Rose and Abby Ussher and Joanne M. Wood. Effects of night-time bicycling visibility aids on vehicle passing distance. Accident Analysis & Prevention. 2020;144:105636. 

8. Pawe l Dro´zdziel and Rafa l Wrona. Problems with Not Recognising the Roadblocks at Reduced Visibility. Transportation Research Procedia. 2020; 44:189- 195. 

9. Noor Elmitiny and Xuedong Yan and Essam Radwan and Chris Russo and Dina Nashar. Classification analysis of driver’s stop/go decision and red-light running violation. Accident Analysis & Prevention. 2010;42(1):101-111. 

10. Teal Evans and Rwth Stuckey and Wendy Macdonald. Young drivers’ perceptions of risk and difficulty: Day versus night. Accident Analysis & Prevention. 2020; 147:105753. 

11. Freed, S. A., Ross, L. A., Gamaldo, A. A., & Stavrinos, D. Use of multilevel modeling to examine variability of distracted driving behavior in naturalistic driving studies. Accident Analysis & Prevention. 2021;152(4):105986. 

12. F. Freuli and G. De Cet and M. Gastaldi and F. Orsini and M. Tagliabue and R. Rossi and G. Vidotto. Cross-cultural perspective of driving style in young adults: 4

Psychometric evaluation through the analysis of the Multidimensional Driving Style Inventory. Transportation Research Part F: Traffic Psychology and Behaviour. 2020; 73:425-432. 

13. Gazis, D., & Maradudin, H. A. The problem of the amber signal light in traffic flow. Operations Research. 1960;8(1):112-132. 

14. Han, I., & Yang, K. S. Characteristic analysis for cognition of dangerous driving using automobile black boxes. International Journal of Automotive Technology. 2009;10(5):597-605. 

15. Md. Mazharul Haque and Amanda D. Ohlhauser and Simon Washington and Linda Ng Boyle. Decisions and actions of distracted drivers at the onset of yellow lights. Accident Analysis & Prevention. 2016; 96:290-299. 

16. Qinaat Hussain and Wael K.M. Alhajyaseen and Kris Brijs and Ali Pirdavani and Tom Brijs. Innovative countermeasures for red light running prevention at signalized intersections: A driving simulator study. Accident Analysis & Prevention. 2020; 134:105349. 

17. Hussain, Q., Alhajyaseen, W., Pirdavani, A., Reinolsmann, N., Brijs, K., & Brijs, T. Speed perception and actual speed in a driving simulator and real-world: a validation study. Transportation Research Part F: Psychology and Behaviour. 2019; 62:637-650. 

18. Motonori Ishibashi and Masayuki Okuwa and Shun’ichi Doi and Motoyuki Akamatsu. Indices for characterizing driving style and their relevance to car following behavior. SICE Annual Conference. 2007:1132-1137. 

19. Jose-Luis Padilla and Candida Castro and Pablo Doncel and Orit Taubman - Ben -Ari. Adaptation of the multidimensional driving styles inventory for Spanish drivers: Convergent and predictive validity evidence for detecting safe and unsafe driving styles. Accident Analysis & Prevention. 2020; 136:105413. 

20. Joanne M. Wood and Gillian Isoardi and Alex Black and Ian Cowling. Night-time driving visibility associated with LED streetlight dimming. Accident Analysis & Prevention. 2018; 121:295-300. 

21. Johnell O. Brooks and Richard R. Goodenough and Matthew C. Crisler and Nathan D. Klein and Rebecca L. Alley and Beatrice L. Koon and William C. Logan and Jennifer H. Ogle and Richard A. Tyrrell and Rebekkah F. Wills. Simulator sickness during driving simulation studies. Accident Analysis & Prevention. 2010;42(3):788-796. 

22. Michael D. Keall and William J. Frith and Tui L. Patterson. The contribution of alcohol to night time crash risk and other risks of night driving. Accident Analysis& Prevention. 2005;37(5):816-824. 

23. Andras Kemeny and Francesco Panerai. Evaluating perception in driving simulation experiments. Trends in Cognitive Sciences. 2003;7(1):31-37. 

24. Max Kinateder and Brittany Comunale and William H. Warren. Exit choice in an emergency evacuation scenario is influenced by exit familiarity and neighbor behavior. Safety Science. 2018; 106:170-175. 

25. Kinateder, Max and Ronchi, Enrico and Nilsson, Daniel and Kobes, Margrethe and M¨uller, Mathias and Pauli, Paul and M¨uhlberger, Andreas. Virtual reality for fire evacuation research. Federated Conference on Computer Science and Information Systems. 2014:313-321. 

26. Panos Konstantopoulos and Peter Chapman and David Crundall. Driver’s visual attention as a function of driving experience and visibility. Using a driving simulator to explore drivers’ eye movements in day, night and rain driving. Accident Analysis & Prevention. 2010;42(3):827-834. 

27. Lee and John. Handbook of Driving Simulation for Engineering, Medicine, and Psychology. Handbook of Driving Simulation for Engineering, Medicine, and Psychology. 2011. 

28. Jing Lin and Lijun Cao and Nan Li. How the completeness of spatial knowledge influences the evacuation behavior of passengers in metro stations: A VR-based experimental study. Automation in Construction. 2020; 113:103136. 

29. Llopis-Castello, D., Camacho-Torregrosa, F. J., Marin-Morales, J., Perez- Zuriaga, A. M., Garcia, A., & Dols, J. F. Validation of a low-cost driving simulator based on continuous speed profiles. Transportation Research Record: Journal of the Transportation Research Board. 2016;2602(2026):104-114. 

30. Long Sun & Ruosong Chang. Reliability and validity of the Multidimensional Driving Style Inventory in Chinese drivers. Traffic injury prevention. 2019;20(2):152–157. 

31. Guangquan Lu and Yunpeng Wang and Xinkai Wu and Henry X. Liu. Analysis of yellow-light running at signalized intersections using high-resolution traffic data. Transportation Research Part A: Policy and Practice. 2015; 73:39-52. 

32. Siwei Ma and Xuedong Yan. Examining the efficacy of improved traffic signs and markings at flashing-light-controlled grade crossings based on driving simulation and eye tracking systems. Transportation Research Part F: Traffic Psychology and Behaviour. 2021; 81:173-189. 

33. Peter Mikoski and Gian Zlupko and D. Alfred Owens. Drivers’ assessments of the risks of distraction, poor visibility at night, and safety-related behaviors of themselves and other drivers. Transportation Research Part F: Traffic Psychology and Behaviour. 2019; 62:416-434. 

34. Panagiotis Papaioannou. Driver behaviour, dilemma zone and safety effects at urban signalised intersections in Greece. Accident Analysis & Prevention. 2007;39(1):147-158. 6

35. Nishant Mukund Pawar and Nagendra R. Velaga. Investigating the influence of time pressure on overtaking maneuvers and crash risk. Transportation Research Part F: Traffic Psychology and Behaviour. 2021; 82:268-284. 

36. Praveena Penmetsa and Srinivas S. Pulugurtha. Risk drivers pose to themselves and other drivers by violating traffic rules. Traffic Injury Prevention. 2017;18(1):63 -69. 

37. Qiong Wu and Feng Chen and Guohui Zhang and Xiaoyue Cathy Liu and Hua Wang and Susan M. Bogus. Mixed logit model-based driver injury severity investigations in single- and multi-vehicle crashes on rural two-lane highways. Accident Analysis & Prevention. 2014; 72:105-115. 

38. Anna-Maria Sourelli and Ruth Welsh and Pete Thomas. Objective and perceived risk in overtaking: The impact of driving context. Transportation Research Part F: Traffic Psychology and Behaviour. 2021; 81:190-200. 

39. Steuer, J. Defining virtual reality: dimensions determining telepresence. Journal of Communication. 2010;42(4):73-93. 

40. Long Sun and Liang Cheng and Qi Zhang. The differences in hazard response time and driving styles of violation-involved and violation-free taxi drivers. Transportation Research Part F: Traffic Psychology and Behaviour. 2021; 82:178- 186. 

41. Orit Taubman - Ben-Ari and Vera Skvirsky. The multidimensional driving style inventory a decade later: Review of the literature and re-evaluation of the scale. Accident Analysis & Prevention. 2016; 93:179-188. 

42. Orit Taubman-Ben-Ari and Mario Mikulincer and Omri Gillath. The multidimensional driving style inventory—scale construct and validation. Accident Analysis & Prevention. 2004;36(3):323-332. 

43. Xinmiao, Fan, Gaofeng, Pan, Yan, & Mao. Investigating the effect of personality on left-turn behaviors in various scenarios to understand the dynamics of driving styles. Traffic injury prevention. 2019;20(8):801-806. 

44. Tamer Yared and Patrick Patterson. The impact of navigation system display size and environmental illumination on young driver mental workload. Transportation Research Part F: Traffic Psychology and Behaviour. 2020; 74:330- 344. 

45. Rainer Zeller and Ann Williamson and Rena Friswell. The effect of sleep-need and time-on-task on driver fatigue. Transportation Research Part F: Traffic Psychology and Behaviour. 2020; 74:15-29.”

Reviewer #1: 

The authors propose that driving style and lighting conditions have an impact on the driver's decision to drive at yellow lights. The study was conducted using a driving simulator and a VR device, which has led to new technological developments to address the driving aspect of the study. The study shows that maladjusted drivers are more likely to run yellow lights and point to relevant solutions that are relevant to solving the problem of yellow light running.

The paper is interesting and should be useful to some practitioners and researchers in the field. However, the main drawback is that the contribution of this work is not specific enough, examining only the two main categories of driving styles without studying them separately in detail. 

I have the following suggestions for how the authors could improve the submission: 

We sincerely thank this reviewer for his/her comments. All comments from this reviewer were carefully addressed in this revised manuscript. The revisions are explained in detail below. 

Introduction: 

1. Although the text is generally well written, there are errors in grammar and misspellings that make comprehension difficult. For example, on page 3 in the paragraph: “modelling- modeling”. For example, in this paper, driving styles or driving style? I strongly recommend that the author revise the text to address these errors prior to publication.

We sincerely thank this reviewer for raising grammatical issues. We have corrected these errors and have thoroughly proofread the entire manuscript, as detailed below:

Page 1, Line 6, 8, 12 :driving styles----driving style 

Page 1, Line 7 : driving styles deserve--driving style deserves 

Page 1, Line 15 :divide--divided 

Page 2, Line 31, 32 : driving styles----driving style 

Page 3, Line 77 : play---- plays 

Page 3, Line 101, 103 : included---- includes 

Page 3, Line 111 : design---- design 

Page 6, Line 186-187 : variables divided ---- variables are divided 

Page 7, Line 225, 226, 228 :driving styles----driving style 

Page 7, Line 227: to the gender ---- to gender 

Page 8, Line 239 : the drivers----those 

Page 8, Line 267 : therefore----so they 

Page 11, Line 384: as dangerous drivers ---- dangerous 

Page 11, Line 395-396: and they trust their skills and believe that no dangerous accidents will occur.---- and trust their skills, no dangerous accidents will occur. 

Page 11, Line 398, 410 :driving styles----driving style 

Page 12, Line 430 : the maladaptive driving styles ---- maladaptive driving styles

2. The traffic running variables in the article include driving speed, distance to the stop line at the start of the yellow light, acceleration noise (or change) before the start of the yellow light,time, and minimum speed. However, the two variables, time, and minimum speed are not shown in the table and decision tree. Are these two variables necessary or unnecessary? If they are not necessary, please fix them, and if they are necessary, please add the relevant data information.

As suggested by the reviewer, we decided to remove the variables time and minimum speed because they were not considered in the study. (Page 8)

3. The ideas presented are already known, so probably you need to improve the discussion so you can add some novelty to this paper.

As suggested by the reviewer, we have improved the Discussion section to make it new and marketable (Page 8-11):

Driving decisions under different light conditions:

“Driver behavior at signalized intersections is considered critical because of its direct impact on traffic safety [36]. And different light conditions may lead to deviations in driving behavior. Such deviations can greatly increase the incidence of traffic accidents. For this reason, VR simulations of lighting scenarios provide realistic and important information for research and are expected to reduce the incidence of deviations.”

“Fig 6 shows that the probability of a driver choosing to pass a yellow light is lower in the daytime situation than in the nighttime situation (31.25% vs 56.25%). This increased probability can be explained by the fact that the driving behavior of the sampled drivers in the nighttime scenario may lead to an increased risk. This result can be attributed to the fact that drivers have different visibility of the road under different lighting conditions [7]. In low visibility conditions, accidents are more likely to occur due to difficulties in recognizing conditions on the road, especially in encounters with pedestrians [8,21]. Conversely, in high visibility conditions, drivers can make the right decision in time to avoid an accident when encountering road conditions.”

“Overall, the findings suggest that drivers tend to choose higher speeds to pass the yellow light in the night scenario than in the day scenario, while the distance to the stop line is shorter and acceleration is higher.”

Driving decisions under different driving styles:

“The results of the study show that driving style has a significant influence on the driver's driving decision to choose to pass a yellow light. This result suggests that driving style has an important basis for drivers to make driving decisions. Driving style predicts driver behavior on the road, which in turn reduces the incidence of traffic accidents.”

“From Fig 6, it can be concluded that non-adaptive drivers are more likely to choose to pass a yellow light than adaptive drivers (59.375% > 28.125%). This can be attributed to the fact that non-adaptive drivers are more likely to make dangerous decisions and tend to violate safe driving norms by passing yellow lights quickly. Adaptive drivers are more likely to make rational decisions due to their emotional stability and tend to follow safe driving guidelines by stopping and waiting at the stop line. This result further emphasizes the importance of driving style in predicting driving behavior.”

“In the literature, drivers' speed is repeatedly cited as a contributing factor in their decision to start at a yellow light. In general, drivers usually choose a faster speed to run a yellow light. To test whether our data would yield factually correct results, an ANOVA was chosen for testing. The results showed that driving style had a significant effect on drivers' driving speeds (F (1,62) = 6.83, p<0.05). Drivers with different driving styles chose different driving speeds. The results also suggest that when non-adaptive drivers drive at higher speeds, they tend to choose a more dangerous route (frequent lane changes) to get through yellow lights. This result can be explained by the fact that non- adaptive drivers tend to perceive themselves as skilled drivers and that higher speeds produce a sense of excitement, which makes non-adaptive drivers feel happy emotions. In contrast, adaptive drivers are ruled by caution and prudence when driving and tend to drive within the prescribed speed limit. As a result, non-adaptive drivers had significantly higher speeds than adaptive drivers (11.53 m/s > 10.13 m/s). Similarly, as acceleration noise increased (F = 0.27, p<0.05) and distance decreased (F = 0.25, p<0.05), the probability of passing a yellow light was significantly higher for non-adaptive drivers compared to adaptive drivers. A similar explanation can be made for the relationship with the probability of passing a yellow light, i.e., the driver's driving style had a significant effect on the change in acceleration, with a positive correlation (F = 0.27, p<0.05). Therefore, to reduce the probability of yellow light jumping by maladjusted drivers, relevant driving regulations should be developed to regulate the behavior of this group of drivers and reduce the probability of traffic accidents.”

The impact of interactions on driving decisions:

“The presence of interaction effects suggests a degree of complexity between driver decision-making at yellow lights and data on speed, distance to the stop line, acceleration, lighting conditions, and driving style. These relationships are further complicated by the effect of driving style on drivers' speed choices at signalized intersections. To test for differential risk, the probability of a driver passing a yellow light was plotted as shown in Figure 6. The results show that the interaction between driving style and lighting conditions had a significant effect (p<0.05) on the driver's driving decision to choose to pass the yellow light. In daytime conditions, maladjusted drivers were more likely to choose to pass the yellow light than adapted drivers (43.75% > 12.5%) and in nighttime conditions (81.25% > 43.75%).”

“The results also suggest that the interaction between lighting conditions and driving style has a negative impact on driving decisions. The effects of different lighting conditions were different for drivers with different driving styles. Firstly, for non- adaptive drivers, adequate lighting mitigated the behavior of passing yellow lights and reduced the likelihood of injury or death (43.75% < 81.25%). Secondly, for adaptive drivers, the effect of light conditions on passing yellow lights was significant (F (1, 31) =0.04, p<0.05), indicating that drivers were more likely to choose to pass yellow lights in nighttime conditions (12.5% < 32.5%).”

Driver’s Gender:

“The probability of passing a yellow light increases with speed for all drivers in both conditions, and males are shorter from the stop line than females.”

“And Fig 5 indicates that there is a gender difference in driving speed, with male drivers driving at a higher speed than female drivers. This result also confirms the influence of light conditions and driving style on driving decisions. That is, daytime adaptive drivers tend to drive at lower speeds and avoid running yellow lights.”

Driver experience:

“Fig 5, 7 and 9 show the probability of passing a yellow light for all driving experience groups. It can be observed that in both cases, the probability of passing the yellow light increases with increasing speed for all drivers, while in the night scenario driving conditions, the probability of passing the yellow light is higher for experienced drivers and the driving distance from the stop line is also greater than for inexperienced drivers. This is because they tend to cross the intersection by increasing their speed at the start of the yellow light. However, as found in this study, the occurrence of this risky behavior can be reduced by predictive information on driving style.”

“Inexperienced drivers appeared to have a lower propensity to pass through yellow lights."

4. The experimental design was demonstrated clearly in the paper. The authors tried to analyze the effects of different demographic information but is it confounded with other variables during group assignment?

We are grateful to the reviewers for pointing this out. We designed a 2 (driving style: adaptive, maladaptive) x 2 (light conditions: day, night) experiment by first surveying drivers for demographic-related information, then dividing drivers' driving styles by their scores on the MDSI questionnaire, and then placing the 2 divided groups of driving styles under different light conditions. We analyzed the different demographics in the context mentioned earlier and did not consider demographic information in the group assignment, thus avoiding confusion with it and variables that could invalidate the results of the analysis. More specifically, one of the aims of our study was to examine the effects of light conditions and driving style on the decision to drive at a yellow light. Demographic variables, however, are only subsidiary to this research purpose and are not the subject of the study. The demographic information considered in this paper is under the independent variable of driving style, and in order to prevent the influence of driving style and lighting conditions, the same number of drivers with the same amount of driving gender and driving experience were selected for the study so that no confounding would occur. For a more thorough explanation, in the experimental section we have added a sub-section on "Driving demographics" to explain in detail the choice of driving experience and driving gender as demographic variables. (Page 4):

“Driving demographics:

Driving gender and driving experience have a significant impact on driving behaviors. Drivers of different genders exhibit different driving behaviors. Male drivers tend to drive faster and are more prone to reckless driving, while female drivers tend to choose to drive more cautiously. Driving experience also has a varying degree of influence on driving behaviors. Experienced drivers tend to choose more dangerous driving styles, while inexperienced drivers tend to be overly cautious and cause anxiety. This paper therefore investigates the differences between the driving styles of drivers in different light conditions by selecting driving gender and driving experience as demographic variables to fill in the gaps in the research. To prevent confounding of demographic variables by driving style and light conditions, drivers of the same driving gender and number of driving experiences were selected for the study.”

5. The authors adopted ANOVA for statistical analysis but lack interpretation. Please add the relevant information in Data Processing.

We agree with the reviewer that an explanation of ANOVA is needed. We have therefore added the relevant details in the Data Processing section.

“The study used analysis of variance to test the experimental hypothesis. Namely, that driving style is significantly associated with yellow light driving decisions. Illumination conditions influence driving decisions across driving styles.” (Page 6)

“20. Jmw, A., Gi, B., Ab, A., & Ic, C. Night-time driving visibility associated with led streetlight dimming - sciencedirect. Accident Analysis & Prevention. 2018; 121: 295-300.”

“34. Panagiotis Papaioannou. Driver behaviour, dilemma zone and safety effects at urban signalised intersections in Greece. Accident Analysis & Prevention. 2007;39(1):147-158.”

6. How were the driving styles manipulated? Please add relevant information in the Experiments section.

We have added the MDSI subsection to the Experimental Equipment section, which focuses on explaining more details about manoeuvring driving styles. (Page 4):

“MDSI:

Based on statements of feelings, thoughts, and behaviors while driving, drivers complete a Likert scale. The scale has a total score of 6, ranging from 1 (not at all) to 6 (very much). As the Cronbach's alpha was reasonable for the four dimensions (0.82 for dangerous driving, 0.82 for anxious driving, 0.77 for angry driving, and 0.70 for cautious driving), each driver's responses to the relevant scales were averaged to produce scores of each of the four driving styles, with higher scores indicating higher levels of that style.”

Reviewer #2:

The authors use a driving simulator and a VR device to study the influence of driving style and lighting conditions on drivers' yellow light decisions. The results show that maladjusted drivers are more likely to run yellow lights. The results of this study are relevant to solving practical problems. The manuscript is well structured and readable. Below I point out some issues I consider the authors need to address for the manuscript to be accepted.

We sincerely thank this reviewer for his/her comments. All comments from this reviewer were carefully addressed in this revised manuscript. The revisions are explained in detail below.

1. There are grammatical errors. Please have a native speaker proofread the manuscript and correct the English grammar and exposition accordingly.

We sincerely thank this reviewer for raising grammatical issues. We have thoroughly proofread the entire manuscript, as detailed below:

Page 1, Line 6, 8, 12 :driving styles----driving style 

Page 1, Line 7 : driving styles deserve--driving style deserves 

Page 1, Line 15 :divide--divided 

Page 2, Line 31, 32 : driving styles----driving style 

Page 3, Line 77 : play---- plays 

Page 3, Line 101, 103 : included---- includes 

Page 3, Line 111 : design---- design 

Page 6, Line 186-187 : variables divided ---- variables are divided 

Page 7, Line 225, 226, 228 :driving styles----driving style 

Page 7, Line 227: to the gender ---- to gender 

Page 8, Line 239 : the drivers----those 

Page 8, Line 267 : therefore----so they 

Page 11, Line 384: as dangerous drivers ---- dangerous 

Page 11, Line 395-396: and they trust their skills and believe that no dangerous accidents will occur.---- and trust their skills, no dangerous accidents will occur. 

Page 11, Line 398, 410 :driving styles----driving style 

Page 12, Line 430 : the maladaptive driving styles ---- maladaptive driving styles

2. The bar graphs in the article are not sufficiently aesthetically pleasing. The visibility and accuracy of the bar graphs are low, so please use professional software to create them.

As suggested by the reviewer, we redrew the relevant graphics to make them vectorially visible.:

specifically as the response to reviewers.

3. An explanation of the demographic variables and why driving experience and driving gender were chosen as demographic variables should be added. Without the explanation of the demographic variables, the significance of the variables is not clear. You can provide details in the Experiments or Results section.

As suggested by the reviewer, in the experimental section we have added a sub-section on "Driving demographics" to explain in detail the choice of driving experience and driving gender as demographic variables. (Page 4):

“Driving demographics:

Driving gender and driving experience have a significant impact on driving behaviors. Drivers of different genders exhibit different driving behaviors. Male drivers tend to drive faster and are more prone to reckless driving, while female drivers tend to choose to drive more cautiously. Driving experience also has a varying degree of influence on driving behaviors. Experienced drivers tend to choose more dangerous driving styles, while inexperienced drivers tend to be overly cautious and cause anxiety. This paper therefore investigates the differences between the driving styles of drivers in different light conditions by selecting driving gender and driving experience as demographic variables to fill in the gaps in the research. To prevent confounding of demographic variables by driving style and light conditions, drivers of the same driving gender and number of driving experiences were selected for the study.”

4. The introduction section is detailed, but the logic between sentences needs strengthening. A systematic and complete revision of the relationships between statements is recommended.

As suggested by the reviewer, we strengthen the logic of the statement in the introductory section (Pages 1-3):

Page 1, Paragraph 2: “Driving style has an important influence on driver decision making. Driving style refers to a driver's habits in terms of speed selection, following distance, tendency to overtake other vehicles and violation of traffic rules [18]. It plays an important role in predicting driving behaviors and reflecting the driver's internal state [43]. Thus, it is meaningful to explore driving style. Today, there are many instruments that can be used to measure driving style, but in this study, we chose to use the Multidimensional Driving Style Inventory (MDSI). Because the MDSI has been shown to be a valid and reliable indicator for assessing driver styles [19,30,41,42]. Generally, MDSI can be divided into four driving styles through eight factors: danger, anger, anxiety, and caution [42]. In detail, dangerous driving style is the one in which the driver deliberately violates driving regulations and is driving in pursuit of excitement, speed, or illegal overtaking; anxious driving is driving in a manner where the driver develops feelings and emotions of alertness and nervousness, accompanied by distracting behaviors; an angry driving style is the one in which the driver displays irritating, angry, and hostile attitudes and behaviors; and cautious driving style refers to safe and cautious driving behaviors [42].”

Page 2, Paragraph 4: “Additionally, light conditions also have an important influence on driver decisions. Firstly, drivers have different hazard perceptions under different lighting conditions. In the night scenario, the driver's hazard perception sensitivity index all but drops, and even a complete lack of awareness of the hazard occurs, which leads to an increased rate of vehicle crashes and a significant increase in the severity of injuries that do not occur in the daytime scenario [4,10,37]. This may be related to the psychological needs of the driver. This is because the psychological needs of drivers are higher in night scenes than in day scenes [4]. Not only that, but the psychological needs of drivers differed between driving styles as well. Therefore, it is relevant to study the influence of lighting conditions on driving style. Although there have been many studies on driving performance under different lighting conditions, such as distracted driving [11,44], sleep driving [6,45], driving risk [21,38], visibility [26,33], and visual attention [26], there is a gap in the research on the driving decisions of drivers with different driving styles under different lighting conditions.”

Page 2, Paragraph 5: “Although there are many studies on signalized intersections, there is still a research gap on the decision-making behaviors of drivers with differing driving styles under different lighting conditions at yellow lights.”

Page 2-3, Paragraph 6: “Early research on driver decision-making at yellow lights has focused on modelling the propensity of drivers to run yellow lights as a function of constructive driving speed, distance from the stop line, and demographic variables such as driver age and gender [10,16,36]. Although early studies on yellow light decision making had significant findings, there were some shortcomings that could not be addressed [1]. Today, with the advancement of technology, many scholars study driving decisions at yellow lights through improved models or functions that better reduce the likelihood of traffic accident development. For example, the inclusion of a connected environment at signalized intersections helps drivers to make safer decisions at the onset of yellow lights [1]. In addition, distracted driving models in the form of mobile phone conversations have been constructed to investigate the impact of distracted driving on yellow light decisions [15]. While these and other related studies have confirmed the importance of drivers' yellow light decisions, it is unclear how the affective information provided by driving style affects drivers' decisions at the signal under different lighting conditions. This research gap prompted the present study.”

Page 3, Paragraph 7: “Driving simulators plays an important role in assessing the human factors of road safety [17,27,29]. However, there are still limitations to the fidelity of driving simulators [25,26]. Therefore, immersive virtual environments (IVEs) based on virtual reality (VR) technology offers an alternative approach to the study of driving behavior. Virtual reality is "a real or simulated environment in which the perceiver experiences a sense of remoteness" [39]. The behavior observed in a virtual reality environment is qualitatively comparable to that of the real world [25].”

Page 3, Paragraph 8: “Today, IVE is an effective research tool with reasonable ecological validity to evoke realistic human driving behavior.”

Page 3, Paragraph 9: “Secondly, what driving decisions does drivers with differing driving styles make under different lighting conditions when the traffic lights are yellow?”

“10. Teal Evans and Rwth Stuckey and Wendy Macdonald. Young drivers’ perceptions of risk and difficulty: Day versus night. Accident Analysis & Prevention. 2020; 147:105753.” 

“15. Md. Mazharul Haque and Amanda D. Ohlhauser and Simon Washington and Linda Ng Boyle. Decisions and actions of distracted drivers at the onset of yellow lights. Accident Analysis & Prevention. 2016; 96:290-299.” 

“16. Qinaat Hussain and Wael K.M. Alhajyaseen and Kris Brijs and Ali Pirdavani and Tom Brijs. Innovative countermeasures for red light running prevention at signalized intersections: A driving simulator study. Accident Analysis & Prevention. 2020;134:105349.” 

“17. Hussain, Q., Alhajyaseen, W., Pirdavani, A., Reinolsmann, N., Brijs, K., & Brijs, T. Speed perception and actual speed in a driving simulator and real-world: a validation study. Transportation Research Part F: Psychology and Behaviour. 2019; 62:637-650.”

“18.Motonori Ishibashi and Masayuki Okuwa and Shun’ichi Doi and Motoyuki Akamatsu. Indices for characterizing driving style and their relevance to car following behavior. SICE Annual Conference. 2007:1132-1137.” 

“19. Jose-Luis Padilla and Candida Castro and Pablo Doncel and Orit Taubman - Ben-Ari. Adaptation of the multidimensional driving styles inventory for Spanish drivers: Convergent and predictive validity evidence for detecting safe and unsafe driving styles. Accident Analysis & Prevention. 2020; 136:105413.” 

“21. Johnell O. Brooks and Richard R. Goodenough and Matthew C. Crisler and Nathan D. Klein and Rebecca L. Alley and Beatrice L. Koon and William C. Logan and Jennifer H. Ogle and Richard A. Tyrrell and Rebekkah F. Wills. Simulator sickness during driving simulation studies. Accident Analysis & Prevention. 2010;42(3):788-796.” 

“22. Keall, M. D., Frith, W. J., & Patterson, T. L. The contribution of alcohol to night time crash risk and other risks of night driving. Accident Analysis & Prevention. 2005; 37(5): 816-824.” 

“23. Kemeny, A., & Panerai, F. Evaluating perception in driving simulation experiments. Trends in Cognitive Sciences. 2003; 7(1): 31-37.” 

“25.Kinateder, Max and Ronchi, Enrico and Nilsson, Daniel and Kobes, Margrethe and Müller, Mathias and Pauli, Paul and Mühlberger, Andreas. Virtual Reality for Fire Evacuation Research. Federated Conference on Computer Science and Information Systems. 2014:313-321.” 

“26. Konstantopoulos, P., Chapman, P., & Crun Da Ll, D. Driver's visual attention as a function of driving experience and visibility. using a driving simulator to explore drivers' eye movements in day, night and rain driving. Accident Analysis & Prevention. 2010; 42(3): 827-834.” 

“27.Lee, & John. Handbook of driving simulation for engineering, medicine, and psychology. Handbook of driving simulation for engineering, medicine, and psychology. 2011.” 

“29. Llopis-Castello, D., Camacho-Torregrosa, F. J., Marin-Morales, J., Perez-Zuriaga, A. M., Garcia, A., & Dols, J. F. Validation of a low-cost driving simulator based on continuous speed profiles. Transportation Research Record: Journal of the Transportation Research Board. 2016; 2602(2026): 104-114.” 

“30. Long Sun & Ruosong Chang. Reliability and validity of the Multidimensional Driving Style Inventory in Chinese drivers. Traffic injury prevention. 2019; 20(2): 152–157.” 

“33. Mikoski, P., Zlupko, G., & Owens, D. A. Drivers' assessments of the risks of distraction, poor visibility at night, and safety-related behaviors of themselves and other drivers. Transportation Research Part F: Traffic Psychology and Behaviour. 2019; 62: 416-434.” 

“34. Papaioannou, P. Driver behaviour, dilemma zone and safety effects at urban signalised intersections in greece. Accident Analysis & Prevention. 2007; 39(1): 147-158.”

“36. Praveena Penmetsa and Srinivas S. Pulugurtha. Risk drivers pose to themselves and other drivers by violating traffic rules. Traffic Injury Prevention. 2017;18(1):63-69.” 

“37. Qiong Wu and Feng Chen and Guohui Zhang and Xiaoyue Cathy Liu and Hua Wang and Susan M. Bogus. Mixed logit model-based driver injury severity investigations in single- and multi-vehicle crashes on rural two-lane highways. Accident Analysis & Prevention. 2014; 72:105-115.” 

“38. Anna-Maria Sourelli and Ruth Welsh and Pete Thomas. Objective and perceived risk in overtaking: The impact of driving context. Transportation Research Part F: Traffic Psychology and Behaviour. 2021; 81:190-200.” “39. Steuer, J. Defining virtual reality: dimensions determining telepresence. Journal of Communication. 2010;42(4):73-93.” 

“41. Orit Taubman - Ben-Ari and Vera Skvirsky. The multidimensional driving style inventory a decade later: Review of the literature and re-evaluation of the scale. Accident Analysis & Prevention. 2016; 93:179-188.” 

“42. Orit Taubman-Ben-Ari and Mario Mikulincer and Omri Gillath. The multidimensional driving style inventory—scale construct and validation. Accident Analysis & Prevention. 2004;36(3):323-332.” 

“43. Xinmiao, Fan, Gaofeng, Pan, Yan, & Mao. Investigating the effect of personality on left-turn behaviors in various scenarios to understand the dynamics of driving styles. Traffic injury prevention. 2019;20(8):801-806.” 

“44. Tamer Yared and Patrick Patterson. The impact of navigation system display size and environmental illumination on young driver mental workload. Transportation Research Part F: Traffic Psychology and Behaviour. 2020; 74:330-344.” 

“45. Rainer Zeller and Ann Williamson and Rena Friswell. The effect of sleep-need and time-on-task on driver fatigue. Transportation Research Part F: Traffic Psychology and Behaviour. 2020; 74:15-29.”

5. There are problems with the format of the references. For example, “Lee, & John. Handbook of driving simulation for engineering, medicine, and psychology. Crc Press. 2011.”

We sincerely thank the editors for raising the issue of references. We have thoroughly proofread the entire manuscript, as detailed below: 

“1. Ali, Y. and Haque, M. M. and Zheng, Z. and Bliemer, Mcj. Stop or go decisions at the onset of yellow light in a connected environment: A hybrid approach of decision tree and panel mixed logit model. Analytic Methods in Accident Research. 2021;31(7–8):100165. 

2. Yasir Ali and Zuduo Zheng and Md. Mazharul Haque and Mehmet Yildirimoglu and Simon Washington Understanding the discretionary lane-changing behaviour in the connected environment. Accident Analysis & Prevention. 2020;137:105463. 

3. Almutairi, O. and Wei, H. Effect of speed/red-light cameras and traffic signal countdown timers on dilemma zone determination at pre-timed signalized intersections. Accident Analysis & Prevention. 2021;154:106076. 

4. Asadamraji, Morteza and Saffarzadeh, Mahmood and Ross, Veerle and Borujerdian, Aminmirza and Sheikholeslami, Sina. A novel driver hazard perception sensitivity model based on drivers’ characteristics: A simulator study. Traffic Injury Prevention. 2019;20(5):492-497. 

5. Francesco Bella. Driving simulator for speed research on two-lane rural roads. Accident Analysis & Prevention. 2008;40(3):1078-1087. 

6. Nipjyoti Bharadwaj and Praveen Edara and Carlos Sun. Sleep disorders and risk of traffic crashes: A naturalistic driving study analysis. Safety Science. 2021;140:105295. 

7. Alex A. Black and Rebecca Duff and Madeline Hutchinson and Ingrid Ng and Kirby Phillips and Katelyn Rose and Abby Ussher and Joanne M. Wood. Effects of night-time bicycling visibility aids on vehicle passing distance. Accident Analysis & Prevention. 2020;144:105636. 

8. Pawe l Dro´zdziel and Rafa l Wrona. Problems with Not Recognising the Roadblocks at Reduced Visibility. Transportation Research Procedia. 2020; 44:189- 195. 

9. Noor Elmitiny and Xuedong Yan and Essam Radwan and Chris Russo and Dina Nashar. Classification analysis of driver’s stop/go decision and red-light running violation. Accident Analysis & Prevention. 2010;42(1):101-111. 

10. Teal Evans and Rwth Stuckey and Wendy Macdonald. Young drivers’ perceptions of risk and difficulty: Day versus night. Accident Analysis & Prevention. 2020; 147:105753. 

11. Freed, S. A., Ross, L. A., Gamaldo, A. A., & Stavrinos, D. Use of multilevel modeling to examine variability of distracted driving behavior in naturalistic driving studies. Accident Analysis & Prevention. 2021;152(4):105986. 

12. F. Freuli and G. De Cet and M. Gastaldi and F. Orsini and M. Tagliabue and R. Rossi and G. Vidotto. Cross-cultural perspective of driving style in young adults: 4

Psychometric evaluation through the analysis of the Multidimensional Driving Style Inventory. Transportation Research Part F: Traffic Psychology and Behaviour. 2020; 73:425-432. 

13. Gazis, D., & Maradudin, H. A. The problem of the amber signal light in traffic flow. Operations Research. 1960;8(1):112-132. 

14. Han, I., & Yang, K. S. Characteristic analysis for cognition of dangerous driving using automobile black boxes. International Journal of Automotive Technology. 2009;10(5):597-605. 

15. Md. Mazharul Haque and Amanda D. Ohlhauser and Simon Washington and Linda Ng Boyle. Decisions and actions of distracted drivers at the onset of yellow lights. Accident Analysis & Prevention. 2016; 96:290-299. 

16. Qinaat Hussain and Wael K.M. Alhajyaseen and Kris Brijs and Ali Pirdavani and Tom Brijs. Innovative countermeasures for red light running prevention at signalized intersections: A driving simulator study. Accident Analysis & Prevention. 2020; 134:105349. 

17. Hussain, Q., Alhajyaseen, W., Pirdavani, A., Reinolsmann, N., Brijs, K., & Brijs, T. Speed perception and actual speed in a driving simulator and real-world: a validation study. Transportation Research Part F: Psychology and Behaviour. 2019; 62:637-650. 

18. Motonori Ishibashi and Masayuki Okuwa and Shun’ichi Doi and Motoyuki Akamatsu. Indices for characterizing driving style and their relevance to car following behavior. SICE Annual Conference. 2007:1132-1137. 

19. Jose-Luis Padilla and Candida Castro and Pablo Doncel and Orit Taubman - Ben -Ari. Adaptation of the multidimensional driving styles inventory for Spanish drivers: Convergent and predictive validity evidence for detecting safe and unsafe driving styles. Accident Analysis & Prevention. 2020; 136:105413. 

20. Joanne M. Wood and Gillian Isoardi and Alex Black and Ian Cowling. Night-time driving visibility associated with LED streetlight dimming. Accident Analysis & Prevention. 2018; 121:295-300. 

21. Johnell O. Brooks and Richard R. Goodenough and Matthew C. Crisler and Nathan D. Klein and Rebecca L. Alley and Beatrice L. Koon and William C. Logan and Jennifer H. Ogle and Richard A. Tyrrell and Rebekkah F. Wills. Simulator sickness during driving simulation studies. Accident Analysis & Prevention. 2010;42(3):788-796. 

22. Michael D. Keall and William J. Frith and Tui L. Patterson. The contribution of alcohol to night time crash risk and other risks of night driving. Accident Analysis& Prevention. 2005;37(5):816-824. 

23. Andras Kemeny and Francesco Panerai. Evaluating perception in driving simulation experiments. Trends in Cognitive Sciences. 2003;7(1):31-37. 

24. Max Kinateder and Brittany Comunale and William H. Warren. Exit choice in an emergency evacuation scenario is influenced by exit familiarity and neighbor behavior. Safety Science. 2018; 106:170-175. 

25. Kinateder, Max and Ronchi, Enrico and Nilsson, Daniel and Kobes, Margrethe and M¨uller, Mathias and Pauli, Paul and M¨uhlberger, Andreas. Virtual reality for fire evacuation research. Federated Conference on Computer Science and Information Systems. 2014:313-321. 

26. Panos Konstantopoulos and Peter Chapman and David Crundall. Driver’s visual attention as a function of driving experience and visibility. Using a driving simulator to explore drivers’ eye movements in day, night and rain driving. Accident Analysis & Prevention. 2010;42(3):827-834. 

27. Lee and John. Handbook of Driving Simulation for Engineering, Medicine, and Psychology. Handbook of Driving Simulation for Engineering, Medicine, and Psychology. 2011. 

28. Jing Lin and Lijun Cao and Nan Li. How the completeness of spatial knowledge influences the evacuation behavior of passengers in metro stations: A VR-based experimental study. Automation in Construction. 2020; 113:103136. 

29. Llopis-Castello, D., Camacho-Torregrosa, F. J., Marin-Morales, J., Perez- Zuriaga, A. M., Garcia, A., & Dols, J. F. Validation of a low-cost driving simulator based on continuous speed profiles. Transportation Research Record: Journal of the Transportation Research Board. 2016;2602(2026):104-114. 

30. Long Sun & Ruosong Chang. Reliability and validity of the Multidimensional Driving Style Inventory in Chinese drivers. Traffic injury prevention. 2019;20(2):152–157. 

31. Guangquan Lu and Yunpeng Wang and Xinkai Wu and Henry X. Liu. Analysis of yellow-light running at signalized intersections using high-resolution traffic data. Transportation Research Part A: Policy and Practice. 2015; 73:39-52. 

32. Siwei Ma and Xuedong Yan. Examining the efficacy of improved traffic signs and markings at flashing-light-controlled grade crossings based on driving simulation and eye tracking systems. Transportation Research Part F: Traffic Psychology and Behaviour. 2021; 81:173-189. 

33. Peter Mikoski and Gian Zlupko and D. Alfred Owens. Drivers’ assessments of the risks of distraction, poor visibility at night, and safety-related behaviors of themselves and other drivers. Transportation Research Part F: Traffic Psychology and Behaviour. 2019; 62:416-434. 

34. Panagiotis Papaioannou. Driver behaviour, dilemma zone and safety effects at urban signalised intersections in Greece. Accident Analysis & Prevention. 2007;39(1):147-158. 6

35. Nishant Mukund Pawar and Nagendra R. Velaga. Investigating the influence of time pressure on overtaking maneuvers and crash risk. Transportation Research Part F: Traffic Psychology and Behaviour. 2021; 82:268-284. 

36. Praveena Penmetsa and Srinivas S. Pulugurtha. Risk drivers pose to themselves and other drivers by violating traffic rules. Traffic Injury Prevention. 2017;18(1):63 -69. 

37. Qiong Wu and Feng Chen and Guohui Zhang and Xiaoyue Cathy Liu and Hua Wang and Susan M. Bogus. Mixed logit model-based driver injury severity investigations in single- and multi-vehicle crashes on rural two-lane highways. Accident Analysis & Prevention. 2014; 72:105-115. 

38. Anna-Maria Sourelli and Ruth Welsh and Pete Thomas. Objective and perceived risk in overtaking: The impact of driving context. Transportation Research Part F: Traffic Psychology and Behaviour. 2021; 81:190-200. 

39. Steuer, J. Defining virtual reality: dimensions determining telepresence. Journal of Communication. 2010;42(4):73-93. 

40. Long Sun and Liang Cheng and Qi Zhang. The differences in hazard response time and driving styles of violation-involved and violation-free taxi drivers. Transportation Research Part F: Traffic Psychology and Behaviour. 2021; 82:178- 186. 

41. Orit Taubman - Ben-Ari and Vera Skvirsky. The multidimensional driving style inventory a decade later: Review of the literature and re-evaluation of the scale. Accident Analysis & Prevention. 2016; 93:179-188. 

42. Orit Taubman-Ben-Ari and Mario Mikulincer and Omri Gillath. The multidimensional driving style inventory—scale construct and validation. Accident Analysis & Prevention. 2004;36(3):323-332. 

43. Xinmiao, Fan, Gaofeng, Pan, Yan, & Mao. Investigating the effect of personality on left-turn behaviors in various scenarios to understand the dynamics of driving styles. Traffic injury prevention. 2019;20(8):801-806. 

44. Tamer Yared and Patrick Patterson. The impact of navigation system display size and environmental illumination on young driver mental workload. Transportation Research Part F: Traffic Psychology and Behaviour. 2020; 74:330- 344. 

45. Rainer Zeller and Ann Williamson and Rena Friswell. The effect of sleep-need and time-on-task on driver fatigue. Transportation Research Part F: Traffic Psychology and Behaviour. 2020; 74:15-29.”

---

## [Decision Letter · Decision Letter 1]

28 Feb 2022

Yellow light decision based on driving style: day or night?

PONE-D-21-39684R1

Dear Dr. Mao,

We’re pleased to inform you that your manuscript has been judged scientifically suitable for publication and will be formally accepted for publication once it meets all outstanding technical requirements.

Kind regards,

Feng Chen

Academic Editor

PLOS ONE

Additional Editor Comments (optional):

Reviewers' comments:

Reviewer's Responses to Questions

**Comments to the Author**

1. If the authors have adequately addressed your comments raised in a previous round of review and you feel that this manuscript is now acceptable for publication, you may indicate that here to bypass the “Comments to the Author” section, enter your conflict of interest statement in the “Confidential to Editor” section, and submit your "Accept" recommendation.

Reviewer #1: All comments have been addressed

Reviewer #2: All comments have been addressed

2. Is the manuscript technically sound, and do the data support the conclusions?

Reviewer #1: Yes

Reviewer #2: Yes

3. Has the statistical analysis been performed appropriately and rigorously? 

Reviewer #1: Yes

Reviewer #2: Yes

4. Have the authors made all data underlying the findings in their manuscript fully available?

Reviewer #1: Yes

Reviewer #2: Yes

5. Is the manuscript presented in an intelligible fashion and written in standard English?

Reviewer #1: Yes

Reviewer #2: Yes

6. Review Comments to the Author

Reviewer #1: (No Response)

Reviewer #2: The manuscript has been revised as per reviewers comments and its current version is ready to publish.

7. PLOS authors have the option to publish the peer review history of their article (what does this mean?). If published, this will include your full peer review and any attached files.

Reviewer #1: No

Reviewer #2: No

---

## [Editor Report · Acceptance letter]

5 Mar 2022

PONE-D-21-39684R1 

Yellow light decision based on driving style: day or night ? 

Dear Dr. Mao:

I'm pleased to inform you that your manuscript has been deemed suitable for publication in PLOS ONE. Congratulations! Your manuscript is now with our production department. 

Kind regards, 

on behalf of

Dr. Feng Chen 

Academic Editor

PLOS ONE